# Inherent Safety Analysis and Sustainability Evaluation of a Vaccine Production Topology in North-East Colombia

Ángel Darío González-Delgado [1,*] 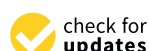, Janet B. García-Martínez [2] and Andrés F. Barajas-Solano [2]

1 Nanomaterials and Computer Aided Process Engineering Research Group (NIPAC), Chemical Engineering Department, University of Cartagena, Cartagena de Indias 130015, Colombia
2 Department of Environmental Sciences, Universidad Francisco de Paula Santander, Av. Gran Colombia No. 12E-96, Cucuta 540003, Colombia
* Correspondence: agonzalezd1@unicartagena.edu.co

**Abstract:** Influenza is a respiratory disease that may cause severe consequences to human health. Influenza caused between 99,000 and 200,000 deaths worldwide in 2019. Studies have reported the presence of this virus in Santander, Colombia, a region with a high humanitarian flow. An influenza vaccine production plant topology has been proposed previously. Nevertheless, the inherent safety and sustainably behavior of this topology is unknown. Process safety plays a crucial role in the evaluation of emerging technologies since it allows the identification of potential risks. Moreover, the current sustainability policies enforce the assessment of processes considering economic, social, and environmental aspects. For this reason, a safety and sustainability evaluation of a vaccine production topology is performed in this work. The inherent safety index (ISI) methodology was implemented to analyze the process. The sustainability evaluation was performed using the sustainability weighted return on investment metric (SWROIM), in which return on investment (ROI), output potential environmental impact (PEI output), total safety inherent index (ITI), and exergy efficiency were considered. The results showed that influenza vaccine production is inherently safe since the total inherent safety index was 11. The destroyed exergy was 378.69 MJ/h, the return on investment was 86%, and the SWROIM was estimated at 81%, which means slightly negative impacts on sustainability.

**Keywords:** sustainability indicators; SWROIM; total inherent safety index; exergy efficiency; influenza vaccine; ROI; potential environmental impacts

## 1. Introduction

Influenza is a respiratory illness produced by the influenza A or B virus. This disease may affect various organs such as the heart, brain, and lungs [1]. In 2019, studies estimated 99,000–200,000 deaths worldwide associated with influenza infections [2]. Karlsoon et al. [3] reported the prevalence of the influenza virus in animals in the Los Llanos region (Colombia). They emphasized the importance of controlling influenza in Colombia since it is a small country with many species. Studies have also revealed the presence of this virus in humans. Studies regarding the virus associated with respiratory infections in children under five years in Santander (Colombia) show that 10.6% of the cases are influenza-positive, according to Garcia-Corzo et al. [4]. This suggests that it is crucial to control influenza spreading. Gasparini et al. [5] present vaccination as the best way to mitigate influenza virus effects. Recently, Contreras-Ropero et al. [6] reported the first simulation of an influenza vaccine plant designed in north-east Colombia to provide enough doses for the region. However, that work is limited to the process simulation and an economic evaluation of a base case, and does not consider the economic indicators based on a techno-economic sensitivity analysis, the inherent safety indicators, the exergy efficiency, the potential environmental impacts, or the sustainability-based return on investment using the SWROIM metric for the process. Hence, the safety and sustainability evaluation of

vaccine production is performed in this work. The inherent safety index (ISI) methodology is used to estimate the safety degree of the process. On the other hand, sustainability is evaluated using the sustainability weighted return on investment metric (SWROIM), which considers economic, energy, environmental, and safety criteria. This work's novelty involves identifying the process's risks and proving a sustainability indicator that considers economic, environmental, energy, and safety aspects.

Safety analysis is relevant for designing more reliable processes [7]. The safety analysis allows for the identification of critical aspects to incorporate actions that mitigate the risks. Rasoulli et al. [8] identified the pharmaceutical industry hazards, finding that the most severe threat is exposure to toxic gases. Meramo-Hurtado et al. [9] performed the inherent safety analysis of the production of chitosan microbeads modified with $TiO_2$ nanoparticles, revealing that the main risks were associated with chemical substance handling. Moreno-Sader et al. evaluated chitosan production from shrimp exoskeletons from a safety viewpoint using ISI methodology. The results showed that the considered process was inherently unsafe [10].

Energy efficiency and consumption must be considered for sustainability. Klemes et al. [11] reported that 3.34 kWh is required to manufacture one vaccine dose, which is considered a high energy consumption and causes pollution. The above indicates the importance of performing analysis that identifies energy improvements, such as the exergy analysis. Dias et al. [12] studied the steam reforming process of glycerol from an exergy viewpoint finding that the exergy efficiency was 75.8%. Kang et al. [13] evaluated the sustainability and energy performance of space and water heating in China during winter and summer. The findings revealed aspects of improving the energy efficiency in space and water heating, such as energy consumption in gas furnaces.

The economic aspect is a determining factor in sustainably. Moreover, economic evaluation is crucial during the process design stage since it shows the profitability of a project. Herrera-Rodriguez et al. [14] performed the economic evaluation of industrial agar production from red algae, and the total capital investment (*TCI*) of the project was estimated at USD 11,937,958.17. Zuorro et al. [15] performed an economic analysis of shrimp biorefinery, finding that the project's return on investment was 65.88%.

Finally, sustainability also includes environmental performance, which can be evaluated through an environmental assessment. This allows the identification of improvements to minimize the project's impact. Cassiani-Cassiani et al. [16] studied the agar production from macroalgae *Gracilaria* sp. from an environmental point of view and reported values higher than 400 PEI/h of PEI output. Herrera-Aristizabal et al. [17] estimated PEI output values higher than 20 PEI/h for a palm-based biorefinery. On the other hand, nowadays, policies have encouraged the industrial sector to analyze the processes from a sustainable viewpoint. The definition of sustainability includes economic, environmental, and social dimensions [18]. Meramo-Hurtado et al. [19] evaluated the sustainability of a lignocellulosic multi-feedstock biorefinery using the sustainability weighted return on investment metric (SWROIM), finding that this indicator was higher compared to the return on investment (ROI) of the project.

## 2. Materials and Methods

### 2.1. Inherent Safety Evaluation

The process of inherent safety is related to the intrinsic properties of an operation due to its nature. The inherent safety index ($I_{TI}$) allows for the measurement of the safety of a process in the conceptual design. This may be calculated through Equation (1), where $I_{CI}$ corresponds to the chemical safety index, and $I_{PI}$ represents the inherent safety index of the processes.

$$I_{TI} = I_{CI} + I_{PI} \tag{1}$$

The chemical safety index represents the chemical features that affect the inherent safety of processes. These features are reactivity, flammability, explosiveness, toxic exposure, and corrosion. This index may be estimated using Equation (2), where $I_{RM,max}$ and $I_{RS,max}$

correspond to the maximum values of the assigned indices for main and side reaction heat. $I_{INT,max}$ describes undesired reactions that may take place between the substances involved in the process. The flammability, toxic exposure, and explosiveness indices are assigned for each substance and $(I_{FL} + I_{EX} + I_{TOX})_{max}$ is the maximum sum of these indices. Finally, $I_{COR,max}$ corresponds to the corrosiveness index assigned according to the construction material required to decrease the corrosiveness rate.

$$I_{CI} = I_{RM,max} + I_{RS,max} + I_{INT,max} + (I_{FL} + I_{EX} + I_{TOX})_{max} + I_{COR,max} \tag{2}$$

The process inherent safety index is related to operating conditions and equipment involved in the process. This index may be estimated using Equation (3), where $I_I$ is the subindex of the inventory, which is assigned according to the amount of mass retained in equipment for 1 h, $I_{T,max}$ and $I_{P,max}$ are the subindex for the maximum process temperature and pressure, respectively. $I_{EQ,max}$ corresponds to the subindex for equipment safety which is determined according to the equipment involved in the process, and $I_{ST,max}$ represents the subindex for safe process structure which is assigned according to the available information about the safety of the technology used [20]. In Table 1, the score ranges for each subindex are shown.

$$I_{PI} = I_I + I_{T,max} + I_{P,max} + I_{EQ,max} + I_{ST,max} \tag{3}$$

**Table 1.** The score ranges for inherent safety subindexes, adapted from [19].

| $I_{CI}$ | Symbols | Score |
|---|---|---|
| Chemical reactivity (main reaction) | $I_{RM,max}$ | 0-4 |
| Chemical reactivity (side reactions) | $I_{RS,max}$ | 0-4 |
| Chemical interactions | $I_{INT,max}$ | 0-4 |
| Flammability | $I_{FL}$ | 0-4 |
| Explosiveness | $I_{EX}$ | 0-4 |
| Toxic exposure | $I_{TOX}$ | 0-6 |
| Corrosiveness | $I_{COR,max}$ | 0-2 |
| **$I_{PI}$** | **Symbols** | **Score** |
| Inventory | $I_I$ | 0-5 |
| Temperature | $I_{T,max}$ | 0-4 |
| Pressure | $I_{P,max}$ | 0-4 |
| Equipment safety | $I_{EQ,max}$ | 0-4 (ISBL); 0-3 (OSBL) |
| Safe process structure | $I_{ST,max}$ | 0-5 |

### 2.2. Sustainability Evaluation

Most process improvements are performed to satisfy economic aspects. The sustainability evaluation also considers energy, environmental, and safety parameters. To estimate a sustainability value, SWROIM is used, which may be calculated through Equation (4).

$$SWROIM = \frac{AEP\left[1 + \sum_{i=1}^{Nindicators} w_i \left(\frac{Indicator_i}{Indicator_i^{Target}}\right)\right]}{TCI} \tag{4}$$

where *TCI* corresponds to the project's total capital investment, *AEP* is the annual net profit of the project, $w_i$ corresponds to the weighting factors of sustainability indicator *i*, *Indicator* and $Indicator_i^{Target}$ are the indicator of the project and the target indicator, respectively. The project designer decides the assignation of the weighting factors for each parameter, considering the information provided in Table 2 [19]. The indicators used in this methodology are described in the following sections. The weighting factor establishes the importance of the sustainability indicator in comparison with the economic indicator. The weighting factor may present values higher or lower than 1, depending on the preference

of the sustainability indicator, considering that values equal to 1 have the same importance as the economic aspect in the SWROIM calculation. Thus, the sum of all weighting factors does not have to be equal to 1.

**Table 2.** Meaning of weighting factor values.

| Value of Weighting Factor | Meaning |
| --- | --- |
| $w_i < 1$ | Lower relevance than economic parameter |
| $w_i = 1$ | Equal relevance as an economic parameter |
| $w_i > 1$ | Higher relevance than economic parameter |

### 2.3. Exergy Indicators

The methodology used for calculating energy indicators was the exergy analysis, which is based on thermodynamics' first and second laws. Exergy analysis provides information about thermodynamic inefficiencies of a process [21]. This tool identifies critical stages in the process; therefore, energy improvements may be proposed. Global exergy efficiency is an essential indicator in this analysis and was used by the authors as a parameter for calculating the SWROIM. Exergy can be defined as the maximum amount of work that may be performed by a system when it is brought into equilibrium with a reference environment [22]. It can be estimated through Equation (5), where $Ex_{total-entry}$ is the exergy associated with the inlet mass flow and utilities; on the other hand, $Ex_{destroyed}$ is the difference between exergy inlet and exergy of product.

$$\eta_{energy} = 1 - \left( \frac{Ex_{destroyed}}{Ex_{total-entry}} \right) \tag{5}$$

### 2.4. Economic Indicators

Techno-economic evaluation provides vital indicators that determine the project feasibility [23]. Return on investment (*ROI*) is an indicator that measures how much profit a company produces from its capital investments [24], which is described in Equation (6). Total capital investment (*TCI*) is the money needed for purchasing and installing equipment. Annual profit after taxes can be calculated as the difference between yearly project sales and total product cost, including operating costs.

$$\%ROI = \frac{Annual\ profit}{TCI} \times 100 \tag{6}$$

### 2.5. Environmental Indicators

The WAR algorithm is a methodology to evaluate processes from an environmental viewpoint based on the potential environmental impacts (PEI). These are defined as the effect caused by chemical substances if they were emitted into the environment. PEI output and PEI generated are considered in this methodology. PEI output can be defined as a measurement of the ability of the process to obtain a product with the minimum discharge of PEI [25] and can be estimated through Equation (7).

$$\hat{i}_{out}^{(t)} = \sum_j^{cp} \dot{M}_j^{(out)} \sum_k X_{kj} \Psi_k + \sum_j^{ep-g} \dot{M}_j^{(out)} \sum_k X_{kj} \Psi_k \tag{7}$$

where $\dot{i}_{we}^{(cp)}$ and $\dot{i}_{we}^{(ep)}$ are the *PEI* associated with residual energy. $\dot{M}_j^{(out)}$ is the mass flow of stream $j$, $X_{kj}$ is the mass fraction of component $k$ in stream $j$, and $\Psi_k$ represents the PEI of component $k$.

Furthermore, the *PEI* reduction percentage by saving energy, estimated using Equation (8), is established to determine the improvement degree of the environmentally improved case compared to the base case.

$$\%R_{energy\ saving}PEI_{output} = \left( \frac{PEI_{output_{max}} - PEI_{output_i}}{PEI_{output_{max}}} \right) * 100 \tag{8}$$

where $PEI_{output_{max}}$ corresponds to the PEI output of the base case and $PEI_{output_i}$ is defined as the PEI output when half of the energy is consumed in the most consumer stage.

## 3. Results

### 3.1. Process Description

Figure 1 shows influenza vaccine production. This process starts with the cell propagation step, where MDCK (Madin–Darby canine kidney) cells grow in a DMEM (Gibco Dulbecco's Modified Eagle Medium) culture. This culture contains amino acids, vitamins, and glucose [26] and it is used to foment the growth of cells. In this stage, 0.39 kg/h of FBS (fetal bovine serum) at 25 °C and 1.01 bar is used as a supplement in the reaction system. Once the desired cell concentration is reached, and the cells are transported to reactors, the ideal cell density for virus propagation is obtained, and 3.56 kg/h of carbon dioxide at 25 °C and 1.01 bar is added to each reactor. Next, in the infection and virus adaptation stage, 0.32 kg/h of FBS are fed at 25 °C. At this stage, 70% of the cells are infected. Then, 3.23 kg/h of flow that contains DMEM, virus, and antibiotics are treated in the milling stage (BM-101), where cell lysis is accelerated to release the virus from the cell. Later, cellular debris is removed in the centrifugation stage, and a flow of 2.35 kg/h is waste that contains DMEM, FBS, and a low amount of antibiotics (CF-101). The virion particles are purified through salt addition in the washing stage (WSH-102). Next, suspended solids and proteins are removed in the microfiltration stage (MF-101), producing 0.38 kg/h of waste. Finally, adjuvants, potassium alum, virus, and penicillin are added in the formulation stage (V-103), and the vials are packed. The influenza vaccine production manufactures 3,083,569,629 units per year with an energy requirement of 161,440 kJ/h. Table 3 includes the pressure, temperature, and mass flow of the main streams involved in the process. A more detailed process description is provided by Contreras-Ropero et al. [6].

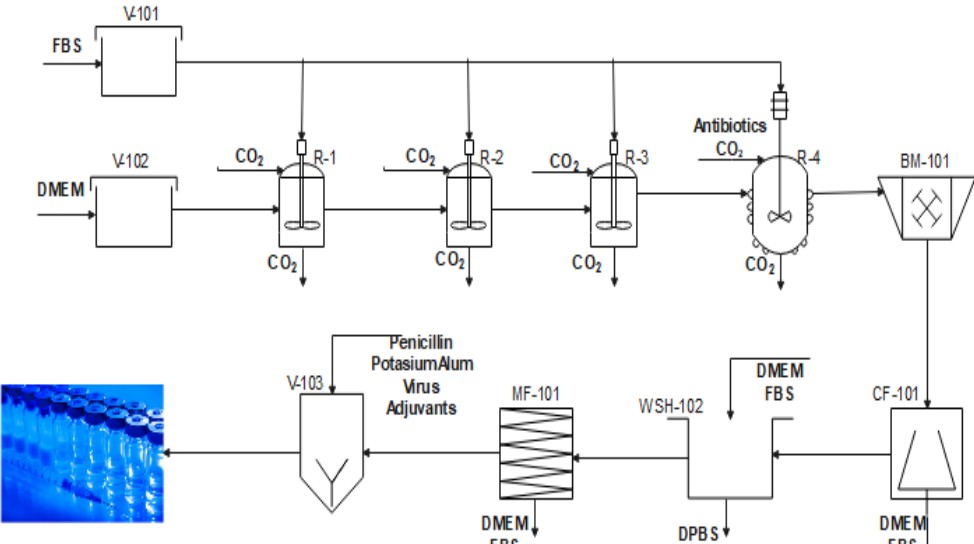

**Figure 1.** Process diagram of the influenza vaccine production in north-east Colombia.

**Table 3.** Pressure, temperature, and mass flow of the main streams of the process.

| Stream | T °C | P (atm) | Mass Flow (kg/h) |
|---|---|---|---|
| Inlet R-1 | 100.00 | 1.00 | 0.19 |
| Outlet R-4 | 39.62 | 1.00 | 3.24 |
| Inlet WSH-102 | 25.00 | 1.00 | 0.89 |
| Inlet MF-101 | 25.00 | 1.00 | 0.76 |
| Outlet MF-101 (waste) | 25.00 | 1.00 | 0.38 |
| Inlet V-103 (main stream) | 25.00 | 1.00 | 0.37 |
| Outlet V-103 (product) | 25.00 | 1.00 | 840.32 |

### *3.2. Inherent Safety Analysis*

#### 3.2.1. Chemical Inherent Safety Index

In Figure 2, the results of the chemical inherent safety index are shown. Pyruvate oxidation takes place in influenza vaccine production, which is slightly exothermic; therefore, the chemical reaction subindex was 1 ($I_{RM, MAX} = 1$). There were no secondary reactions; thus, the chemical secondary response subindex was null ($I_{RS, MAX} = 0$). The chemical interactions subindex was 1 ($I_{INT, MAX} = 1$) due to the formation of inoffensive and non-toxic chemicals. The dangerous chemical substances subindex was calculated, taking into account the information from the safety data sheet of each substance. This subindex was 3 ($I_{(tox, fl, ex)MAX} = 3$), which corresponds to urea due to its toxicity. The construction material of some equipment is stainless carbon to decrease corrosion; therefore, the corrosiveness subindex was 1.

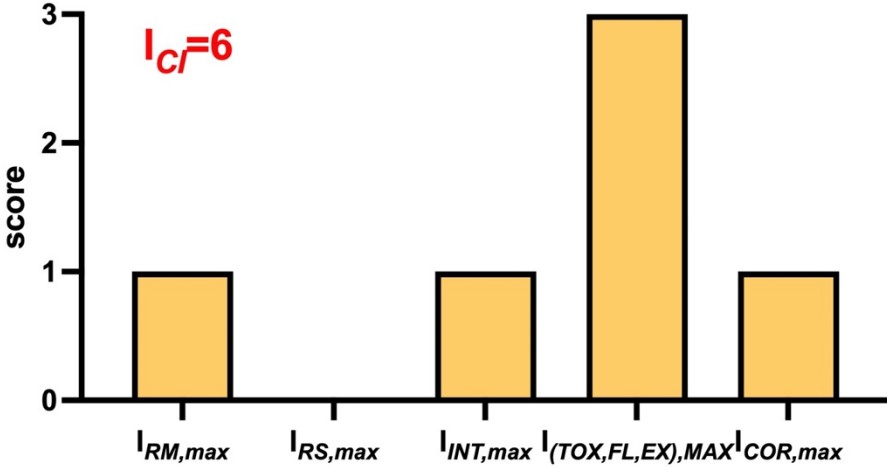

**Figure 2.** Total scores of chemical inherent safety subindexes.

#### 3.2.2. Process Inherent Safety Index

Figure 3 shows the results for the process inherent safety index. The total inventory of the vaccine production was estimated at 0.8 t/h; therefore, the inventory subindex resulted in 0 ($I_I = 0$). The maximum temperature of the process was registered at the infection and virus adaptation stage (111, 54 °C); thus, the score for this subindex was ($I_{T, max} = 1$). The entire process is carried out at atmospheric pressure (1 atm), which is not considered a risk. The score for this subindex resulted in being null ($I_{P, max} = 0$). The score of the equipment safety subindex was 3 ($I_{eq, max} = 3$) since the reactors are the most unsafe equipment in vaccine production. The safe structure index is related to the engineering of the process. Vaccine production through this method has already been implemented, making this process well known and reliable; thus, the score of the safety structure index was 1 ($I_{IST} = 3$).

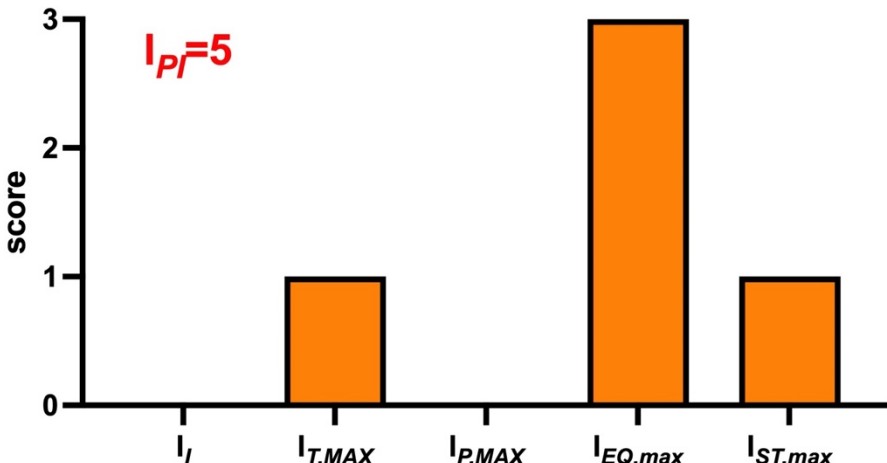

**Figure 3.** Total scores of process inherent safety subindexes.

### 3.2.3. Total Inherent Safety Index

Figure 4 shows the results for the Total Inherent Safety Index ($I_{ISI} = 11$) and the contribution of the chemical and process safety index.

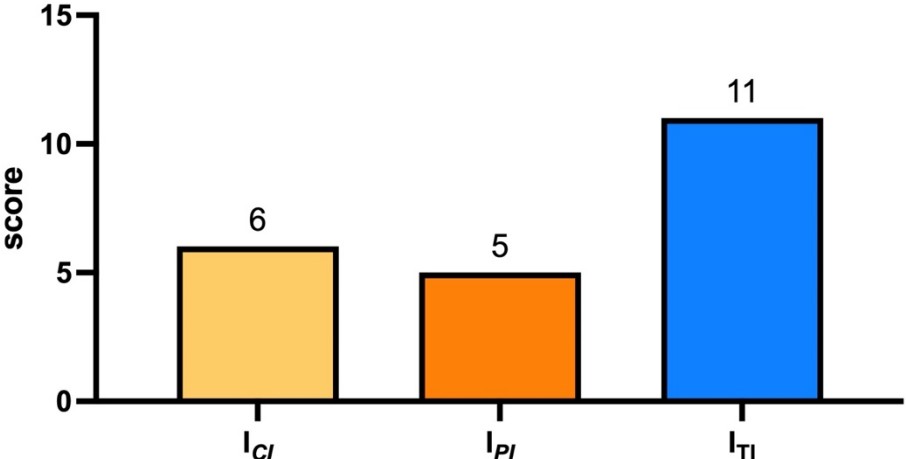

**Figure 4.** Total inherent safety index of influenza vaccine production.

### 3.3. Sustainability Evaluation

The exergy associated with the mass inlet was calculated considering the chemical and physical exergies of the streams involved in the process. The destroyed exergy was estimated as the difference between the exergy mass inlet and the exergies of products. Finally, the efficiency was calculated using Equation (5). The results are shown in Figure 5.

On the other hand, the return on investment was estimated through Equation (6). The annual profit or profit after taxes (PAT) was calculated as the difference between yearly sales and total product cost, including operating costs, considering the taxes. In this study, it was assumed that the construction time of the plant is 3 years, with an income tax rate of 39%, and an interest rate of 9%. The salvage value was considered as 10% of depreciable fixed capital investment. The results of this study are shown in Figure 6.

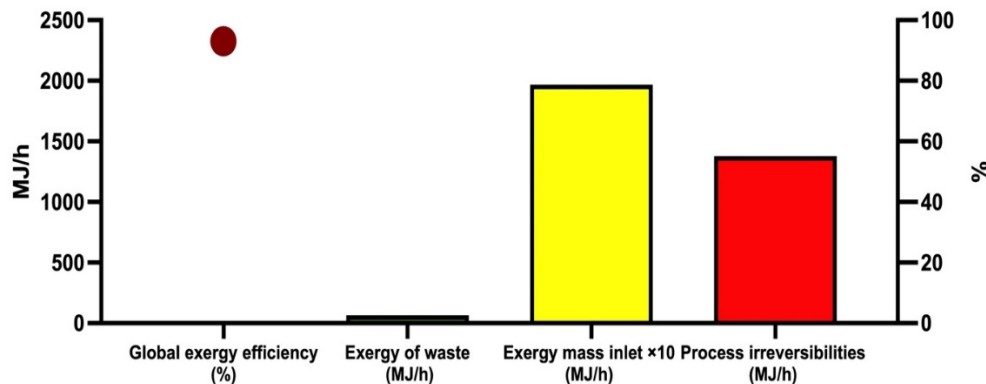

**Figure 5.** Exergy analysis results of influenza vaccine production.

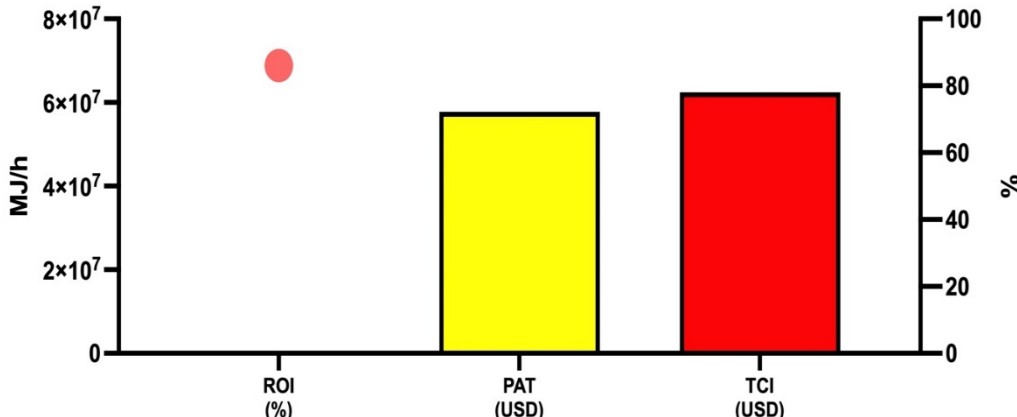

**Figure 6.** Economic evaluation results of influenza vaccine production.

The potential environmental impacts per hour (PEI/h) were estimated in this study. The global mass balance of the process was entered in WARGUI software. Carbon was considered the energy source of the process. Four cases were established to analyze the results; in the first case, neither the product stream nor energy consumption was considered. In the second case, product stream was considered in the outcome, while energy consumption was not considered. Energy consumption was taken into account in the third case. Finally, energy consumption and stream product were considered in the fourth case. Figure 7 shows the result of the environmental assessment.

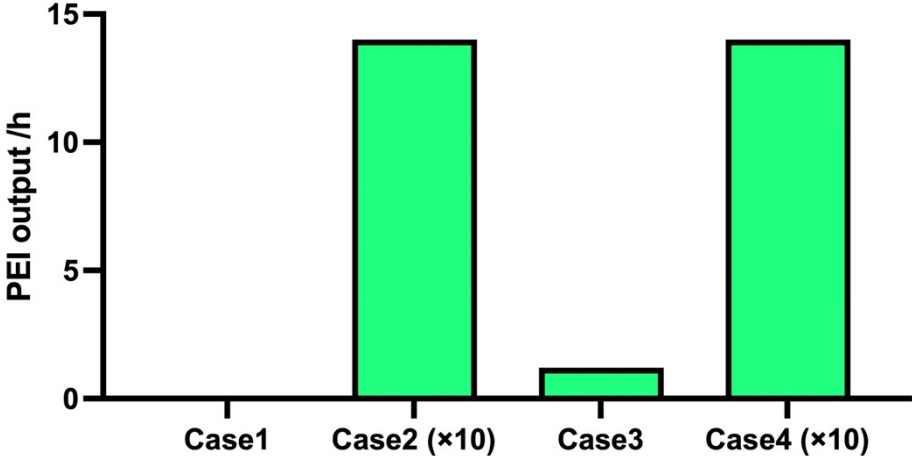

**Figure 7.** Total PEI output of influenza vaccine production.

The value for the energy parameter weighting factor was established at 0.4 ($w_i = 0.4$) following the recommendations disclosed by Meramo-Hurtado et al. [19] for processes with high energy consumption, such as pharmaceutical processes. For the safety parameter, a weighting factor of 0.5 ($w_i = 0.5$) was established due to the plant location near residential zones [19]. Moreover, a high value of ISI negatively affected the process economy. Therefore, the safety target value was considered unfavorable in the SROIWM calculation. Finally, the value for the environmental parameter weighting was set at 0.6 due to the high impact of the pharmaceutical industry [27].

Target indicators ($Indicator^{target}$) must be set for each parameter for SWROIM calculation. The current indicators of the process ($Indicator^i$) must also be set to show the performance. The energy target indicator was set as exergy efficiency of 100%. A total inherent safety index of 24 or lower means a safe process [28]. The ISI obtained in this process was 11. Therefore, this value was set as a target safety indicator. Finally, the environmental target indicator was established as the PEI reduction of 50% due to energy saving. The information for weighting factors, indicators, and target indicators is found in Table 4.

**Table 4.** Weighting factors, indicators, and target indicators for the sustainability evaluation.

| Aspect | Index | $Indicator^i$ | $Indicator^{target}$ | ($w_i$) |
|--------|-------|-----------|----------------|------|
| Energy | Exergy efficiency | 93% | 100% | 0.4 |
| Safety | Total inherent safety index ($I_{TI}$) | 11 | 11 | 0.5 |
| Environmental | $\%R_{saving\ energy}PEI_{output}$ | 5% | 50% | 0.6 |

The SWROIM was estimated at 81%, which is lower than the ROI of the process. Moreover, a sensibility analysis was performed to evaluate the impact of each factor on the sustainability weighted return on investment metric. The values of index weighing were changed in each case, as shown in Table 5; however, the target indicators were reminded equally. Four cases were established. In the first case, the energy aspect had more importance ($w_i = 1$) than the safety and environmental aspects ($w_i = 0.5$). In the second case, economic and environmental were equally relevant ($w_i = 1$), while energy and safety aspects had less relevance ($w_i = 0.5$). In the third case, the economic and safety aspects were more relevant ($w_i = 1$) than the environmental and energy aspects ($w_i = 0.5$). Table 5 summarizes the information for the weighting factors in all cases. Finally, in the fourth case, all parameters had the same relevance ($w_i = 1$). The sensibility analysis results are shown in Figure 8.

**Table 5.** Weighting factors for the study cases.

| | Weighting Factor ($w_i$) | | | | |
|--------|-----------|--------|--------|--------|--------|
| Criteria | Base Case | Case 1 | Case 2 | Case 3 | Case 4 |
| Environmental | 0.4 | 0.5 | 1 | 0.5 | 1 |
| Energy | 0.5 | 1 | 0.5 | 0.5 | 1 |
| Safety | 0.6 | 0.5 | 0.5 | 1 | 1 |

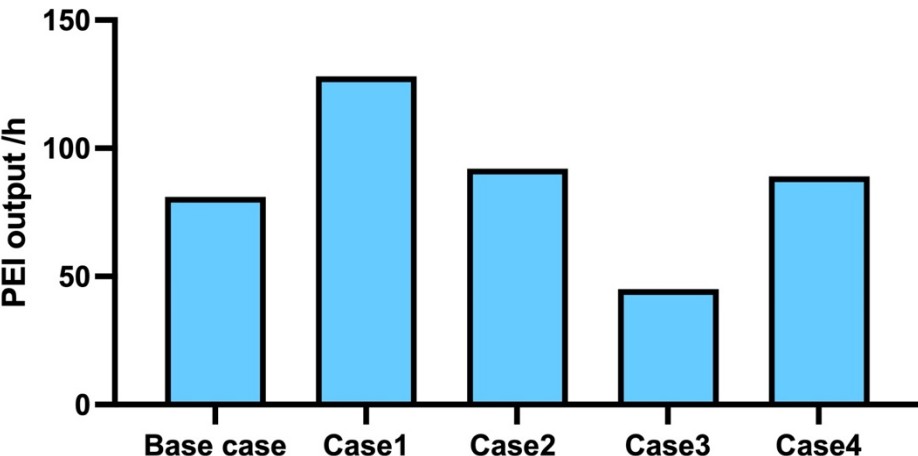

**Figure 8.** SWROIM results for all scenarios evaluated.

## 4. Discussion

According to Figure 2, the dangerous substances subindex contributed the most to the chemical index. Urea was the substance with the maximum dangerous substance subindex due to the threshold limit value (10 ppm) [29]. On the other hand, Figure 3 shows that the equipment safety subindex was the majority contributor to the process safety index because of the reactor presence, which means an equipment safety subindex of 3, according to this methodology developed by [21]. The above implies that precautions should be established in the reactor system to decrease the potential hazards associated with the equipment and increase the safety of the process as runaway reactions, control of the temperature in exothermic reactions, undesired reactions, increases in pressure by the presence of $CO_2$, among others, and to increase the safety of the process. From Figure 4, the chemical substances index is slightly higher than the process safety index, which means chemical substances involved in the process require careful handling. This agrees with the study by Solorzano-Alvarez et al. [30], where the risks involved in producing vaccines for animal use were evaluated. The findings revealed that biological and chemical substance handling is a significant occupational risk. The total inherent safety was estimated at 11, which means the vaccine production is inherently safe since the total inherent safety is lower than the limited value recommended by Luján-Ornelas [18] to consider a chemical process as a secure process.

Concerning sustainability indicators, vaccine production shows a good energy performance since its global exergy efficiency (93%) indicates low energy losses. Zhang [31] evaluated the energy performance in the pharmaceutical industry in Singapore, both in industrial facilities and office buildings, obtaining a low energy efficiency mainly caused by the cooling and ventilation of office buildings, which are not considered in the energy evaluation of influenza vaccine production.

On the other hand, the influenza vaccine production plant needs a higher capital investment than the hepatitis B vaccine plant (USD 3,700,000) mainly due to the higher production capacity of the influenza vaccine plant compared to the hepatitis B vaccine plant (4,000,000 doses per year) [32]. Ledley et al. reported net profits higher than USD 1.9 trillion for large pharmaceutical companies, which are high compared to influenza vaccine production plant profits [33].

Output potential environmental impacts were lower in cases 1 and 3, where the product stream was not considered, indicating that this stream is the largest contributor to PEI output; therefore, the product stream may cause damage to the environment. This agrees with the research conducted by Hasija et al. [34], whose findings revealed that each manufactured dose of mRNA vaccine contributed between 0.1 and 0.2 kg of $CO_2$ equivalent to the footprint. The above confirms the good environmental performance of the vaccine production.

The result of SWROIM is slightly lower than ROI since energy, safety, and environmental aspect were considered. From Figure 8, case 1 presents the highest SWROIM, where economic and energy aspects had the maximum relevance. This revealed the energy parameter was the most relevant aspect of the SWROIM results. Implementing strategies to increase exergy efficiency are recommended, such as decreasing energy consumption in the critical stages. On the other hand, case 3 had the lowest SWROIM, which means the safety parameter did not influence the SWROIM results a lot since the vaccine production is already a safe process. For future research, it is recommended to study other key indicators according to the nature of the process and consider them in the SWROIM calculation. This will provide a broader sustainability analysis.

## 5. Conclusions

In this work, the inherent safety assessment and sustainability evaluation were performed for influenza vaccine production. The total inherent index was estimated at 11, which means the process is inherently safe. The higher subindex corresponds to the dangerous chemical substance's subindex, and hence safe handling of substances involved in the process is recommended. The influenza vaccine production is sustainable considering that it had an SWROIM value of 81%, lower than ROI (86%), which indicates the parameters have a slightly negative impact on sustainability. The safety parameter had the weakest influence on sustainability results, whereas the energy parameter was the stronger determinant. Hence, exergy efficiency should be increased to have better results. For future research, it is recommended to incorporate more parameters that allow for the broader evaluation of sustainability.

**Author Contributions:** Conceptualization, Á.D.G.-D.; methodology, A.F.B.-S., J.B.G.-M. and Á.D.G.-D.; software, Á.D.G.-D.; validation, Á.D.G.-D., J.B.G.-M. and Á.D.G.-D.; formal analysis, A.F.B.-S., J.B.G.-M. and Á.D.G.-D.; investigation, A.F.B.-S., J.B.G.-M. and Á.D.G.-D.; resources, A.F.B.-S., J.B.G.-M. and Á.D.G.-D.; data curation, A.F.B.-S., J.B.G.-M. and Á.D.G.-D.; writing—original draft preparation, A.F.B.-S., J.B.G.-M. and Á.D.G.-D.; writing—review and editing, A.F.B.-S. and Á.D.G.-D.; visualization, Á.D.G.-D.; supervision, Á.D.G.-D.; project administration, Á.D.G.-D.; funding acquisition, Á.D.G.-D. All authors have read and agreed to the published version of the manuscript.

**Funding:** This research received no external funding.

**Institutional Review Board Statement:** Not applicable.

**Informed Consent Statement:** Not applicable.

**Data Availability Statement:** The data used in this study are available on request to the corresponding author.

**Acknowledgments:** The authors thank the University of Cartagena and Universidad Francisco de Paula Santander for providing the equipment required to successfully conclude this study.

**Conflicts of Interest:** The authors declare no conflict of interest.

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
