# Peer review of "Inherent Safety Analysis and Sustainability Evaluation of a Vaccine Production Topology in North-East Colombia"

_sustainability, doi:10.3390/su14169985_

Round 1

Reviewer 1 Report

The present study is quite important as it is representing the safety analysis and sustainability evaluation of a vaccine plant. I am not qualified enough to judge this study but I could say it is well written and easy to follow. However, I have a felling that the author should write the manuscript in past tense instead of present tense (in some place).  This is very common technique that we write our result in past form and compare with the published result in present tense. I would recommend authors to present the manuscript in such a way so that it is very clear to the general reader. 

Abstract needs some numerical value if possible. 

Keywords: I would prefer to use different keywords than the title. Can we add "inherent safety index", "total safety inherent index"

Line 36: Need a reference for this kind of statement

Line 74: ISI already introduced before. Do you still need to mention full form?

I would suggest to revisit the introduction and introduce a paragraph to mention the research gap.

The comment Line 138: Exergy or energy? Please double check

Figure 6: label the right side bar (what does that the percentage mean?). X axis also need labelling. Error bar missing from the bar graph.

Figure 7: Author should provide SD or SE in the bar graph. 

Figure 8: Label Y axix, Add error bar. No need the percentage on the bar.

Line 278: Need reference

Line 281: The authors found the equipment safety subindex is majority contributer, but they should explain what does it mean. 

Line 322: Why the plam oil sustainibility has been compared againest vaccind sustainibility.  I am lost. Please check and compare your result with relevant reference. 

Author Response

We would like to respectfully thank the reviewer 1 for his/her very accurate and eloquent comments, which have enabled us to improve the quality of this article. Below we give detailed answers to all the comments made.

-. Abstract needs some numerical value if possible. 

Thanks for your comment, besides the value of SWROIM (81 %) and total inherent safety index (11), the values of destroyed exergy (1,378.69 MJ/h) and return on investment (86%) were included.

-Keywords: I would prefer to use different keywords than the title. Can we add  “inherent safety” “safety inherent index.”

The keywords were changed in order to use different words than the tittle and “safety inherent index” was added in this section, according to the reviewer’s suggestion.

Line 36: Need a reference for this kind of statement.

Thanks for your comment, Th Garcia-Corzo et al. [4] e reference for the statement mentioned (García-Corzo et al., 2016) was included at the end of the line.

Line 74: ISI already introduced before. Do you still need to mention full form?

Taking into account the suggestion of the reviewer, the full form of ISI was removed from line 74.

I would suggest to revisit the introduction and introduce a paragraph to mention the research gap.

We would like to thank the reviewer for this helpful suggestion, The introduction was reordered to clarify the research gap as follows:

“This suggests that it is crucial to control influenza spreading. Gasparini et al. present vaccination as the best way to mitigate influenza virus effects [5]. In Colombia, there is no company specializing in influenza vaccine production. Therefore, a vaccine production plant in North-East Colombia has been previously proposed by the authors [6], however neither the inherent safety indicators nor the sustainability based return on investment using the SWROIM metric for the process have been calculated. Hence, the safety and sustainability evaluation of vaccine production is performed in this work. The inherent safety index (ISI) methodology is used to estimate the safety degree of the process. On the other hand, sustainability is evaluated using the sustainability weighted return on in-vestment metric (SWROIM), which considers economic, energy, environmental, and safety criteria. This work's novelty consists of identifying the risks of the process and proving a sustainability indicator that considers economic, environmental, energy, and safety aspects”.

The comment Line 138: Exergy or energy? Please double check

Thanks for your comment, the methodology used for calculation of energy indicators was the exergy analysis, and we use the exergy efficiency as parameter for calculation of the SWROIM. Exergy can be defined as the maximum amount of work that may be performed by a system when it is brought into the equilibrium with a reference environment. Taking into account your comment, the section 2.3 was rewritten in order to give more clarity.

Figure 6: label the right-side bar (what does that the percentage mean?). X axis also need labelling. Error bar missing from the bar graph.

Thanks for your suggestion, The right-side bar, which corresponds to the percentage of return on investment, was labeled as well as X axis for PAT and TCI. On the other hand, the data represented in figures 2,3,4,5,6,7 and 8 were obtained from simulation results and environmental, economic, exergy and inherent safety calculation methodologies, therefore values are unique for the base case studied, however, in figure 8 a sensibility analysis was performed for evaluation of SWROIM taking into account different weighting factors.

Figure 7: Author should provide SD or SE in the bar graph. 

The data represented in figures 2,3,4,5,6,7 and 8 were obtained from simulation results and environmental, economic, exergy and inherent safety calculation methodologies, therefore values are unique for the base case studied, however, in figure 8 a sensibility analysis was performed for evaluation of SWROIM taking into account different weighting factors.

Figure 8: Label Y axix, Add error bar. No need the percentage on the bar.

Thanks for your comment, the percentages on the bars were removed and Y axis was labeled. The data represented in figures 2,3,4,5,6,7 and 8 were obtained from simulation results and environmental, economic, exergy and inherent safety calculation methodologies, therefore values are unique for the base case studied, however, in figure 8 a sensibility analysis was performed for evaluation of SWROIM taking into account different weighting factors

Line 278: Need reference

The reference number 29 was added, which describes the threshold limit (TLV) value for urea.

Line 281: The authors found the equipment safety subindex is majority contributor, but they should explain what it means. 

The equipment safety subindex shows the highest contribution to the safety process index (IPI) which means that precautions should be established in the reaction system to decrease the potential hazards associated to the equipment and increase the safety of the process as runaway reaction, control of the temperature in exothermic reactions, undesired reactions, increases in pressure by the presence of CO2, among others. This explanation was added to the Discussion section.

Line 322: Why the palm oil sustainability has been compared against vaccine sustainability.  I am lost. Please check and compare your result with relevant reference. 

Despite of research gap (inexistence of vaccine production sustainability parameters evaluation under the methodologies used in this study), new references (from reference 30 to 34) were added in order to compare the results to other process related to influenza vaccine production

Reviewer 2 Report

The manuscript "Inherent safety analysis and evaluation of the sustainability of a vaccine production plant in northeastern Colombia" makes an evaluation of the sustainability of a vaccine production plant in Colombia. There are several issues to be addressed in this manuscript before its publication. For instance:

- The abstract is difficult to follow. Authors are encouraged to improve it by improving the sequence in which ideas are presented and developed.

- The introduction section needs to be improved: it is difficult to follow, there is no clear order to present the arguments/ideas; the topics of the literature review are mixed without a clear structure; and the research gap is not indicated, so the contribution of the manuscript is not entirely clear.

- Are there any guidelines for the selection of weighting factors?

- How did the authors define PEI_output_i?

- The results section must be introduced by providing details about the process you are evaluating (it is not enough just the description in section 2)

- There are differences between what is shown in Figure 4 and what is said about it. Please explain it deeply.

- How did the authors estimated the exergy and destroyed exdergy? What conditons and fractions were considered in the computations? 

- Why the sum of all weighting factors is not equal to 1?  

Author Response

We would like to respectfully thank the reviewer 2 for his/her very accurate and eloquent comments, which have enabled us to improve the quality of this article. Below we give detailed answers to all the comments made. In the attached file, you can find a more detailed answer for your comments. 

Reviewer 3 Report

Revise citation 17 in the methodology paragraph, which is inconsistent with the context of vaccine production

Author Response

We would like to respectfully thank the reviewer 3 for your comments, which have enabled us to improve the quality of this article. Below we give detailed answers to all the comments made.

Revise citation 17 in the methodology paragraph, which is inconsistent with the context of vaccine production

 Thanks for your comment, it was a typo, Citation 17 (currently citation 26) was corrected.

Reviewer 4 Report

The analysis of safety and sustainability of the production of vaccines is an important resourcer for the development of the regions, which in the future, reduce the impact of public health indicators of infectious diseases, improving the health and well-being of the populations, especially, of the underdeveloped counties. Good initiative! 

Author Response

The analysis of safety and sustainability of the production of vaccines is an important resource for the development of the regions, which in the future, reduce the impact of public health indicators of infectious diseases, improving the health and well-being of the populations, especially, of the underdeveloped counties. Good initiative!

The authors thank to the reviewer for the encouraging comment and hope this study to contribute to the development of safe and sustainable processes in the regions.

Round 2

Reviewer 1 Report

The author have significantly improved the manuscript based on the previous comments.  Few grammatical errors could be fixed during proof reading.

Author Response

Reviewer 1:

We would like to respectfully thank the reviewer 1 for his/her very accurate and eloquent comments, which have enabled us to improve the quality of this article. Below we give detailed answers to all the comments made.

-The author have significantly improved the manuscript based on the previous comments.  Few grammatical errors could be fixed during proof reading. 

Thanks for your comment, the document was revised again using specialized software in order to fix grammatical mistakes.

Reviewer 2 Report

This reviewer thanks to the authors for dealing with ll his concerns. They are encouraged to clearly state the differences between their previous research publication (reference [6]) and the current manuscript.

Author Response

Reviewer 2

We would like to respectfully thank the reviewer 2 for his/her very accurate and eloquent comments, which have enabled us to improve the quality of this article. Below we give detailed answers to all the comments made.

This reviewer thanks to the authors for dealing with ll his concerns. They are encouraged to clearly state the differences between their previous research publication (reference [6]) and the current manuscript.

Thanks for your comment, the differences between reference [6] and the current manuscript were specified in the revised version of the manuscript as follows:

Recently, Contreras-Ropero et al. [6] reported the first simulation of an influenza vaccine plant designed in north-east Colombia to provide enough doses for the region. However, that work is limited to the process simulation and an economic evaluation of a base case, and does not consider the economic indicators based on a technoeconomic sensitivity analysis, the inherent safety indicators, the exergy efficiency, the potential environmental impacts or the sustainability-based return on investment using the SWROIM metric for the process